# Theoretical Nanoarchitectonics of GaN Nanowires for Ultraviolet Irradiation-Dependent Electromechanical Properties

**DOI:** 10.3390/ma16031080

**Published:** 2023-01-26

**Authors:** Kun Yang, Guoshuai Qin, Lei Wang, Minghao Zhao, Chunsheng Lu

**Affiliations:** 1School of Mechanics and Safety Engineering, Zhengzhou University, Zhengzhou 450001, China; 2School of Electromechanical Engineering, Henan University of Technology, Zhengzhou 450001, China; 3Henan Institute of Metrology, Zhengzhou 450001, China; 4School of Mechanical Engineering, Zhengzhou University, Zhengzhou 450001, China; 5Henan Key Engineering Laboratory for Anti-Fatigue Manufacturing Technology, Zhengzhou University, Zhengzhou 450001, China; 6School of Civil and Mechanical Engineering, Curtin University, Perth, WA 6845, Australia

**Keywords:** ultraviolet photoexcitation, electromechanical coupling, photoconductive effects, photothermal effect, GaN nanowires

## Abstract

In this paper, we propose a one-dimensional model that combines photoelectricity, piezoelectricity, and photothermal effects. The influence of ultraviolet light on the electromechanical coupling properties of GaN nanowires is investigated. It is shown that, since the ultraviolet photon energy is larger than the forbidden gap of GaN, the physical fields in a GaN nanowire are sensitive to ultraviolet. The light-induced polarization can change the magnitude and direction of a piezoelectric polarization field caused by a mechanical load. Moreover, a large number of photogenerated carriers under photoexcitation enhance the current density, whilst they shield the Schottky barrier and reduce rectifying characteristics. This provides a new theoretical nanoarchitectonics approach for the contactless performance regulation of nano-GaN devices such as photoelectric sensors and ultraviolet detectors, which can further release their great application potential.

## 1. Introduction

As a direct wide band gap semiconductor material, GaN exhibits high-power density, superior electrical and thermal conductivities, strong radiation resistance, and high breakdown voltage [1]. Owing to the size effect, GaN nanostructures exhibit a smaller Young’s modulus and a higher quality factor, which makes them have obvious advantages in the application of nano-mechanical systems [2,3]. In addition, because of the piezoelectric and semiconductive properties, piezoelectric potential generated in a crystal can effectively regulate the carrier transport capacity of an interface/unction region under mechanical loading [4,5]. Such a unique synergistic effect makes the piezoelectric potential produce a similar “gate circuit”, and hence, many novel modern electromechanical coupling devices have been developed, such as piezoelectric charge-coupled devices [6,7,8] and energy conversion supplies [9,10,11,12,13]. Inevitably, there is an urgent need for the active regulation of device performance.

GaN is a natural optoelectronic material. Due to the fact that the ultraviolet photon energy is larger than the forbidden gap of GaN (the energy band difference between conduction and valence bands [14]), the physical fields in a GaN nanowire are sensitive to ultraviolet. Irradiation can stimulate the generation, separation, transport, and recombination of carriers in GaN [15]. The available studies have mainly focused on regulating the electrical transport characteristics of piezoelectric semiconductors (PSCs) under mechanical loads [16] and doping [17]. There are few reports on regulating their electromechanical properties by means of light excitation [18,19,20,21]. For example, based on the photoelectric experiment of ZnO nanowires, Wang and Zhang found that ultraviolet light can weaken Schottky’s rectification characteristics [22,23,24]. To the best of our knowledge, however, it is still a lack of quantitative analysis from the theoretical and numerical aspects. In practical applications, however, a device is inevitably affected by light irradiation, which can deteriorate the electrical properties of GaN materials through the generation-recombination of carriers. That is, radiation causes a certain disturbance of electrical properties. Therefore, it is necessary to investigate the influence of light irradiation on the electromechanical properties of GaN structures. It is expected that a new approach can be developed for the performance regulation of GaN devices without doping.

In this paper, GaN nanowires under a combined photoexcitation and electrical load are comprehensively investigated by using both theoretical and numerical methods. The paper is organized as follows. In Section 2, a one-dimensional (1-D) thermo-piezoelectric theory is first proposed, including the photoconductive and photothermal effects. Then, in Section 3, the influence of ultraviolet irradiation is analyzed on the physical fields of GaN nanowires, and photoexcitation regulation of the electrical transport properties is discussed in an Ag-GaN Schottky junction. Finally, several main conclusions are summarized in Section 4.

## 2. Basic Equations for a GaN Nanowire under Light Irradiation

Uniform light irradiation can cause temperature rising in a semiconductor structure. According to the principle of heat balance [25], the change in temperature versus time yields
(1)$$C_{T}\frac{d(T-T_{p})}{dt}+G_{T}(T-T_{0})=SI,$$
where *C_T_* is the heat capacity, and *G_T_* = *H*_g_ *A*_g_ represents the heat exchange coefficient, with the air thermal convection coefficient *H*_g_ = 4 W K^−1^ m^−2^ and *A*_g_ the contact area between the semiconductor and air [26]. *T_p_* and *T_0_* are the initial and room temperatures, respectively. *S* denotes the illumination area, and *I* is the light intensity.

At the beginning of illumination, the amount of variation in temperature Δ*T*(0) = 0. Solving Equation (1), the temperature change can be obtained as
(2)$$\Delta T(t)=T-T_{p}=\Delta T_{\mathrm{opt}}(1-e^{-t/\tau_{\theta }}),$$
where Δ*T*_opt_ = *S I/G_T_* is the maximum change of temperature in a steady state, and *τ_θ_* is the thermal time constant. Here, it is worth noting that, when the illumination time t ≫*τ_θ_*, the temperature change Δ*T* (t) = Δ*T*_opt_ = *S I/G_T_*.

The electron-hole pairs in a semiconductor can be excited when the photon energy is higher than the band gap width [22]. That is, GaN can be excited by ultraviolet light to produce non-equilibrium carriers. Under uniform illumination, the non-equilibrium carrier concentrations in a steady state are described as [13]
(3)$$\begin{array}{l} \Delta n_{\mathrm{opt}}=\beta \alpha P_{\mathrm{opt}}\lambda /hce^{-\alpha d}\tau_{n},\\ \Delta p_{\mathrm{opt}}=\beta \alpha P_{\mathrm{opt}}\lambda /hce^{-\alpha d}\tau_{p}, \end{array}$$
where *β* is the internal quantum efficiency, representing the number of photocarrier pairs excited by each photon. *α* is the absorption coefficient, *P*_opt_ is the illumination intensity, *λ* is the wavelength, and *h* and *c* are the Planck constant and light velocity, respectively. *d* represents the incident depth, and *τ_n_* and *τ_p_* are the lifetimes of non-equilibrium electrons and holes, respectively.

As illustrated in Figure 1, let us assume that photoexcited carriers gradually decay with the transmission depth of incident light. In the case of a 1-D nanowire with a radius of *r*, the average concentration of photo-generated carriers on its cross-section can be expressed as
(4)$$\begin{array}{l} \Delta {\bar{n}}_{\mathrm{opt}}=(\iint_{x^{2}+y^{2}\leq r^{2}}\Delta n_{\mathrm{opt}}dxdy)/\pi r^{2},\\ \Delta {\bar{p}}_{\mathrm{opt}}=(\iint_{x^{2}+y^{2}\leq r^{2}}\Delta p_{\mathrm{opt}}dxdy)/\pi r^{2}. \end{array}$$

Taking a 1-D GaN nanorod with a length of 2*L* as an example (see Figure 2), the c-axis is along the z-direction, with a uniform beam of ultraviolet light vertically irradiated on the upper surface. Its physical and mechanical behaviors are governed by the motion equation, electrostatics Gauss’s law, and the current continuity equation [27,28,29,30], that is
(5)$$\begin{array}{l} \frac{\partial \sigma_{zz}}{\partial z}=0,\\ \frac{\partial D_{z}}{\partial z}=q(p-n+N_{D}^{+}-N_{A}^{-}),\\ \frac{\partial J_{z}^{n}}{\partial z}=-qU_{n},\\ \frac{\partial J_{z}^{p}}{\partial z}=qU_{p}, \end{array}$$
where *σ_zz_*, *D_z_*,
$J_{z}^{n}$, and $J_{z}^{p}$ denote the stress tensor, electric displacement, electron concentration density, and hole current density. $N_{D}^{+}$ and $N_{D}^{-}$ represent the ionization degrees of donor and acceptor impurities, respectively. *q* is the unit charge (1.602 × 10^−19^ C), and *n* and *p* are the electron and hole doping concentrations. *U_n_* and *U_p_* are the net recombination rates of free electrons and holes. Here, the generation and recombination of free electrons and holes are in a dynamic equilibrium state, that is, *U_n_* and *U_p_* are equal to 0.

For a 1-D PSC with the polarization direction along the *z*-axis, the constitutive equation in Cartesian coordinates can be written as [31,32,33,34]
(6)$$\begin{array}{l} \sigma_{zz}=c_{33}\varepsilon_{zz}-e_{33}E_{z}-\lambda_{33}\Delta T_{\mathrm{opt}},\\ D_{z}=e_{33}\varepsilon_{zz}+\kappa_{33}E_{z}+p_{33}\Delta T_{\mathrm{opt}},\\ J_{z}^{n}=qn\mu_{33}^{n}E_{z}+qd_{33}^{n}\frac{\partial n}{\partial z},\\ J_{z}^{p}=qp\mu_{33}^{p}E_{z}-qd_{33}^{p}\frac{\partial p}{\partial z}, \end{array}$$
where *ε*_zz_ is the strain tensor, *E_z_* is the electric field strength, *c_33_* and *e_33_* are the elastic and piezoelectric coefficients, *κ*_33_ is the dielectric constant, *λ*_33_ is the thermal expansion coefficient, and *p*_33_ is the pyroelectric coefficient. $\mu_{33}^{n}$ and $\mu_{33}^{p}$ are the mobilities of electrons and holes, and $d_{33}^{n}$ and $d_{3}^{p}$ represent the electron and hole diffusion constants. The mobility and diffusion of free carriers satisfy the Einstein relation [35], namely
(7)$$\frac{\mu_{33}^{n}}{d_{33}^{n}}=\frac{\mu_{33}^{p}}{d_{33}^{p}}=\frac{q}{k_{B}T_{0}}.$$
where *k_B_* is Boltzmann’s constant and *T*_0_ is the reference temperature. The strain *ε*_zz_ and the electric field *E_z_* are related to the mechanical displacement *u* and the electric potential *φ*, respectively, that is
(8)$$\begin{array}{l} \varepsilon_{zz}=\frac{\partial u_{z}}{\partial z},\\ E_{z}=-\frac{\partial \varphi }{\partial z}, \end{array}$$
where *u_z_* and *φ* are the mechanical displacement and electric potential, respectively.

Substituting Equation (6) into Equation (4), the governing equations are obtained by
(9)$$\begin{array}{l} c_{33}\frac{\partial^{2}u}{\partial z^{2}}+e_{33}\frac{\partial^{2}\varphi }{\partial z^{2}}-\lambda_{33}\frac{\partial (\Delta T_{\mathrm{opt}})}{\partial z}=0,\\ e_{33}\frac{\partial^{2}u}{\partial z^{2}}-\kappa_{33}\frac{\partial^{2}\varphi }{\partial z^{2}}+p_{33}\frac{\partial (\Delta T_{\mathrm{opt}})}{\partial z}=q(p-n+N_{D}^{+}-N_{A}^{-}),\\ -qn\mu_{33}^{n}\frac{\partial^{2}\varphi }{\partial z^{2}}+qd_{33}^{n}\frac{\partial^{2}n}{\partial z^{2}}=0,\\ -qp\mu_{33}^{p}\frac{\partial^{2}\varphi }{\partial z^{2}}-qd_{33}^{p}\frac{\partial^{2}p}{\partial z^{2}}=0. \end{array}$$

For an *n*-type GaN nanowire, the concentrations of acceptor and donor impurities are *N + D* = 1 × 10^23^ m^−3^ and $N_{A}^{–}$ = $N_{i}^{2}$/*N + D*, where *N_i_* is the concentration of intrinsic carriers. Other relevant material constants are listed in Table 1 [36,37,38,39,40]. Generally speaking, an analytic solution for such a nonlinear model is difficult to be obtained. Hence, to solve the photoexcitation physical problem, a numerical iterative method is adopted by using the PDE module in COMSOL Multiphysics software. Here it is worth noting that Guo and Yang obtained the approximate analytical solution of 1-D piezoelectric semiconductors by a perturbation method, and in comparison with the results from COMSOL, it can be applied to verify the reliability and accuracy of our calculation [41,42].

## 3. Results and Discussion

Under the ultraviolet light with a wavelength of 350 nm and the mechanical conditions as illustrated in Figure 2, when there is no applied current across (in and out) the nanorod, the boundary conditions at the two ends can be written as
(10)$$\begin{array}{l} \sigma_{z}(\pm L)=f,\\ D_{z}(\pm L)=0,\\ J_{z}^{n}(\pm L)=0,\\ J_{z}^{p}(\pm L)=0. \end{array}$$

Here, *n* and *p* satisfy the following electrical neutral conditions
(11)$$\begin{array}{l} \int_{-l}^{l}\left(p-N_{A}^{-}-\Delta {\bar{p}}_{\mathrm{opt}}^{}\right)dz=0,\\ \int_{-l}^{l}\left(n-N_{D}^{+}-\Delta {\bar{n}}_{\mathrm{opt}}^{}\right)dz=0. \end{array}$$

At the position *z* = 0, the conditions of displacement and potential are
(12)$$\begin{array}{l} u(0)=0,\\ \varphi (0)=0. \end{array}$$

Due to the photoconductive effect, a large number of photogenerated carriers are produced with the increase in light intensity, which obviously enhances the concentration of free electrons and holes (see Figure 3a,b). In addition, ultraviolet light changes the distribution of carriers, especially at both ends. This is attributed to the synergy of piezoelectric and pyroelectric effects. That is, due to the photothermal effect, temperature increases under irradiation of light, which leads to the separation of positive and negative ions in GaN nanowires, resulting in pyroelectric charges. The polarity of pyroelectric polarization is opposite to that of piezoelectric polarization. With the increase of irradiation intensity, pyroelectric polarization becomes dominant, which changes the distribution of piezoelectric polarization charges (see Figure 4a). Similarly, because the direction of pyroelectric potential is opposite to the piezoelectric field generated by pressure, the piezoelectric potential is weakened and the comprehensive potential is even reversed (see Figure 4b,c). Consistent with the potential, the piezoelectric field decreases with the light intensity. When the light intensity reaches a certain value, the pyroelectric field plays a dominant role, and the comprehensive polarization field in GaN is opposite to that without ultraviolet light (see Figure 4d).

Let us take the Ag electrodes at both ends of a GaN nanowire as an example, which forms a double Schottky contact. As illustrated in Figure 2, the boundary conditions are
(13)$$\begin{array}{l} \sigma_{z}(\pm L)=f,\\ V(-L)=V_{a}+V_{bi}+V_{b},\\ V(L)=0,\\ J_{z}^{n}(\pm L)=-qv_{\mathrm{rec}}^{n}(n-n_{m}),\\ J_{z}^{p}(\pm L)=qv_{\mathrm{rec}}^{p}(p-p_{m}), \end{array}$$
where *V_a_* is an applied bias voltage, *V_bi_* is the built-in voltage, and *V_b_* is the piezoelectric potential (i.e., the potential difference between the two ends caused by photoexcitation). $V_{rec}^{n}$ and $V_{rec}^{p}$ denote the thermal recombination velocities of electrons and holes at the Schottky interface, respectively. *n_m_* and *p_m_* are the critical electron and hole concentrations, which can be represented as [43,44,45,46,47]
(14)$$\begin{array}{l} n_{m}=N_{c}e^{-\Phi_{B}/k_{B}T},\\ p_{m}=n_{i}^{2}/n_{m},\\ N_{c}=2{\left(\frac{2\pi m_{e}k_{B}T}{h^{2}}\right)}^{3/2}, \end{array}$$
where *Φ_B_* is the GaN surface barrier when the electron energy is equal to the Fermi level, and *T* is the absolute temperature. *m*_e_ = 1.82 × 10^−31^ kg is the effective mass of conduction band electrons, and *N_c_* = 2.23 × 10^24^ m^−3^ denotes the effective density of states of conduction bands. The built-in voltage, *V_bi_* is defined by [43,44,45]
(15)$$V_{bi}=\Phi_{B}-k_{B}T/q\ln(N_{c}/n_{0}),$$
and the Schottky contacts barrier height *qΦ_B_* can be represented as [43,44]
(16)$$q\Phi_{B}=q\Phi_{M}-q\chi ,$$
which is the difference between the working function of silver (*qΦ_M_* = 4.26 eV) [23] and the electron affinity (*qχ* = 4.1 eV) of GaN [38], leading to the potential difference, *Φ_B_* = 0.16 eV.

It is seen from Figure 5a that ultraviolet light can significantly change the *I-V* characteristics of a GaN Schottky junction. In the absence of light, there is an obvious rectification characteristic due to the Schottky barrier between GaN nanowires and Ag electrode. However, with the increase of ultraviolet power density, the Schottky contact rectification weakens and gradually shifts to the Ohmic contact. This is due to the pyroelectric polarization caused by the photothermal effect, which reduces the barrier height of a junction region. In addition, a large number of carriers generated by photoexcitation have a shielding effect on characteristics of the Schottky rectifier. When the photo-generated carriers increase to a particular concentration, the Schottky barrier is completely shielded, and the rectifying characteristics may be lost. That is, ultraviolet excitation can be applied to regulate the electrical transport behavior of GaN nanodevices.

As shown in Figure 5b, the current increases gradually with the light intensity, regardless of a positive or a negative bias voltage. However, the increasing amplitude becomes smaller and tends to be saturated. That is mainly due to the photoconductivity effect, producing a large number of photo-generated carriers that increases the current density. In addition, the photothermal effect increases with the optical power density and produces a thermal electric field, which enhances the Schottky barrier and the shielding effect of the Schottky junction, and thus, the current is gradually saturated [47,48].

## 4. Conclusions

By using a 1-D PSC model, we have investigated the photoexcitation-dependent electrical behaviors in a GaN nanowire, which are conducive to applications in ultra-fast optics, nonlinear optics, photothermal detection, computational memory, and biocompatibility photodetectors. The main conclusions can be drawn as follows:By incorporating the photoconductive and photothermal effects, a multi-field coupling model is proposed, which can consider photoexcitation nonequilibrium carriers in PSCs.Due to the synergy of piezoelectricity, photoconductivity, and photoexcitation induced-pyroelectricity, the physical field distributions in a GaN nanowire are significantly affected by ultraviolet, including the polarization charge, potential, electric field, and carrier concentration.Ultraviolet light can be applied to regulate the height of the Schottky barrier and even make the rectifying characteristics disappear. That is, the electrical correlation characteristics of a GaN Schottky junction device are highly sensitive to ultraviolet light. This provides a novel, non-contact method for tuning the electrical transport performance of a GaN Schottky junction device.Finally, it is worth noting that, as a preliminary study and for simplification, the effects of ultraviolet excitation on the electromechanical properties of nanowires were only investigated by using theoretical and numerical methods in this work. Obviously, further experimental studies need to be done, such as the tests of the photoconductivity and photothermal effects, to verify the regulation of ultraviolet light on the electrical transmission properties and reveal the physical mechanism of such a complex and coupling problem.

## Figures and Tables

**Figure 1 materials-16-01080-f001:**
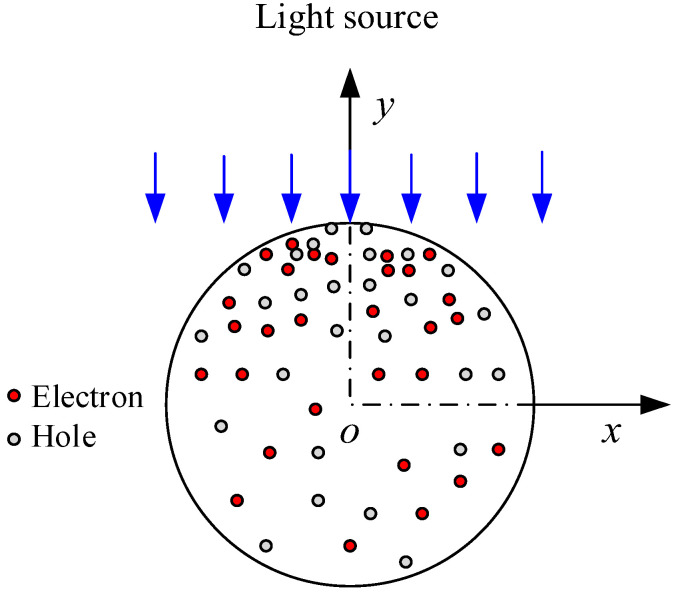
A schematic attenuation diagram of photoexcited carriers with the transmission depth under incident light.

**Figure 2 materials-16-01080-f002:**
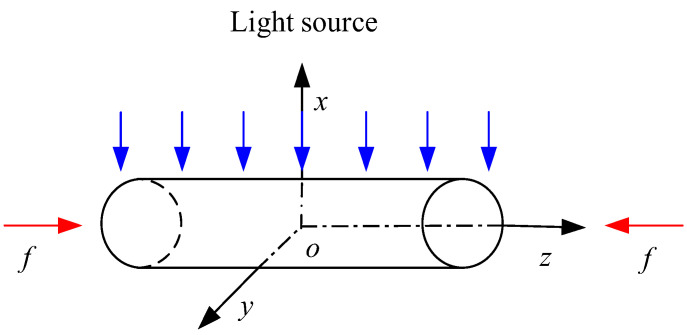
The physical model of ultraviolet irradiation on a GaN nanowire.

**Figure 3 materials-16-01080-f003:**
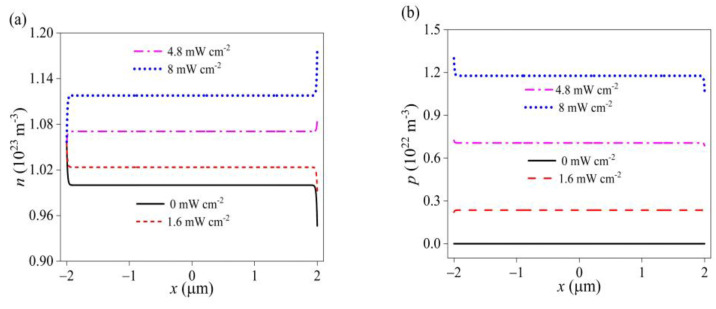
The carrier concentration distribution of (**a**) electron and (**b**) hole with different ultraviolet intensities under a compressive stress of −5 MPa.

**Figure 4 materials-16-01080-f004:**
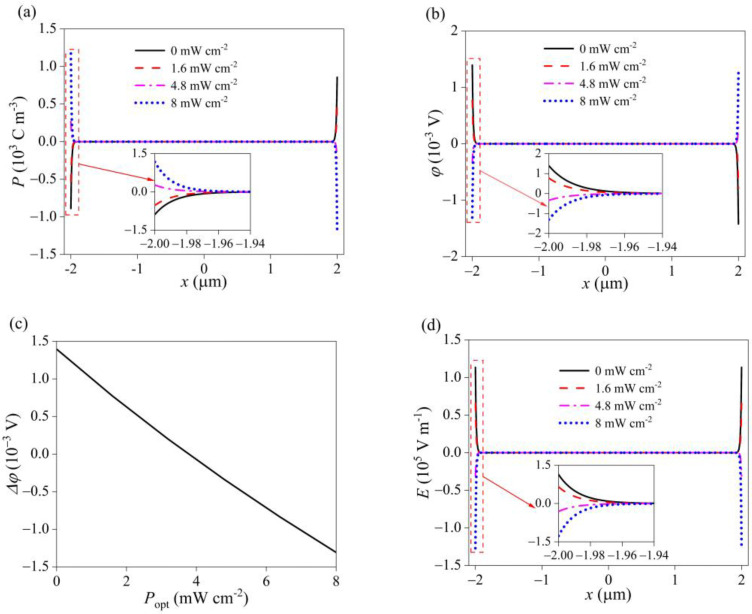
The physical field distributions of (**a**) the polarization charge density, (**b**) the electric potential, (**c**) the variation of the potential at the left end, and (**d**) the electric field under different ultraviolet intensities.

**Figure 5 materials-16-01080-f005:**
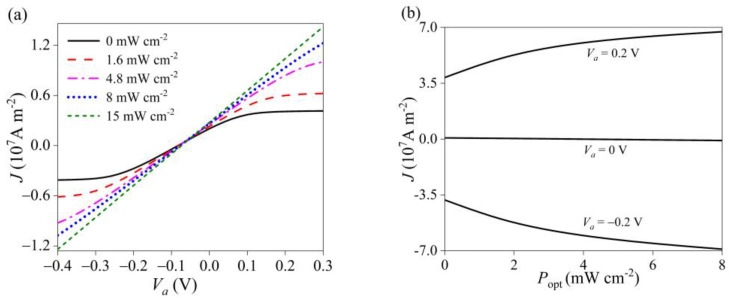
The characteristics of (**a**) *I–V* and (**b**) *I–P*_opt_ in a GaN nanowire under different ultraviolet intensities.

**Table 1 materials-16-01080-t001:** Material constants used in the analysis of a GaN nanowire.

Property	Parameter	Value	Unit
Elastic constant	*c* _33_	289.2	GPa
Piezoelectric constant	*e* _33_	0.61	C m^−2^
Dielectric constant	*κ* _33_	9.39 × 10^−11^	F m^−1^
Hole mobility constant	*μ* p 33	192	cm^2^ V^−1^ s^−1^
Hole diffusion constant	*d* p 33	5	cm^2^ s^−1^
Electron mobility constant	$\mu_{33}^{n}$	560	cm^2^ V^−1^ s^−1^
Electron diffusion constant	$d_{33}^{n}$	25	cm^2^ s^−1^
Thermal expansion coefficient	*λ* _33_	1.69 × 10^6^	N m^−2^ K^−1^
Pyroelectric constant	*p* _33_	−3.8 × 10^−6^	C m^−2^ K^−1^
Intrinsic carrier concentration	*N_i_*	3.43 × 10^−4^	m^−3^
Photogenic electron lifetime	*τ_n_*	1.67 × 10^−5^	s
Photogenic hole lifetime	*τ* _p_	1.67 × 10^−5^	s
Optical absorption coefficient	*α*	1 × 10^5^	cm^−1^
Heat capacity at constant pressure	*C* _p_	490	J kg^−1^ K^−1^

## Data Availability

Data available on request due to restrictions e.g., privacy or ethical. The data presented in this study are available on request from the corresponding author.

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
