# Peer review of "Theoretical Nanoarchitectonics of GaN Nanowires for Ultraviolet Irradiation-Dependent Electromechanical Properties"

_materials, 2023, doi:10.3390/ma16031080_

Round 1

Reviewer 1 Report

Attached.

Author Response

Response to Reviewer 1 Comments

Point 1: The first sentence in the abstract (In this paper, based on a one-dimensional model that links the photoelectricity, piezoelectricity and photothermal effect together, we systematically investigate the influence of ultraviolet light on the electromechanical coupling properties of a GaN piezoelectric semiconductor nanowire.) is very long and confusing; this should be divided into two sentences with a clear understanding.

Response 1: As suggested, the sentence in Abstract has been revised and divided into two sentences (see par. 1 on page 1).

Point 2: The current Abstract looks like a general discussion or introduction. The authors should shift the general discussion to the introduction part and re-write a clear abstract that highlights the main findings of the work and a suitable novelty of the work.

Response 2: Following the suggestion, we have rewritten the Abstract to highlight the main findings and the novelty of the work (see page 1).

Point 3: The authors should provide a clear novelty of the current work and explain the difference between the current and the already available polished similar literature such as https://doi.org/10.1021/nl051860m, https://iopscience.iop.org/article/10.1088/0957-4484/23/16/165701, and their previous work.

Response 3: As suggested, the relevant discussion has been added to clarify the novelty of the current work and its difference from others and our previous work (see pars. 2 and 3 on page 1).

Point 4: What is the forbidden gap of GaN, and how this disturbed the ultraviolet photon energy? This should be mentioned in the introduction with a suitable reference.

Response 4: As suggested, we have added a detail description on the forbidden gap of GaN with relevant references (see par. 3 on page 1).

Point 5: For Figure 5a, it is shown that ultraviolet light can significantly change the I-V characteristics of GaN Schottky junction. But the applied ultraviolet light with an short intensity of 8 mW cm-2 is not enough to conclude this statement. The author should at least apply an ultraviolet light of intensity 15 mW cm-2, which is a bit standard for various Schottky junctions.

Response 5: Have done as suggested (see new Figure 5a on page 8).

Point 6: The conclusion is also a simple discussion, this should be completely modified based on the outcomes and applied techniques.

Response 6: Following the suggestion, we have rewritten the section Conclusions based on the outcomes and applied techniques (see pars. 3-6 on page 9).

Reviewer 2 Report

This manuscript actually provides interesting data. These data are valuable of being published in well-authorized scientific journals such as Materials. However, the provided data sets are somehow unbalanced and insufficient. Such points have to be fixed by revisions. Please see below.

1) This manuscript seams to be theoretical and calculation work. However, it is not clear from the current title, whether experimental work or calculation work. It is better to change the title more clearly. In addition, inclusion of one conceptual term for materials and nano is recommended to make impression more innovative. I may suggest use of an emerging concept, nanoarchitectonics (as post-nanotechnology concept, see https://pubs.rsc.org/en/content/articlelanding/2021/nh/d0nh00680g). For example, the title like ... Theoretical nanoarchitectonics of GaN nanowires for ultraviolet irradiation-dependent electromechanical properties ... may sound more innovative and clear.

2) In conclusion, instead of only fact lists, more description including future perspectives had better be added.

3) Comparisons over the past approach have to be made to emphasize novelty and superiority of the current approach.

Author Response

Response to Reviewer 2 Comments

Point 1: This manuscript seams to be theoretical and calculation work. However, it is not clear from the current title, whether experimental work or calculation work. It is better to change the title more clearly. In addition, inclusion of one conceptual term for materials and nano is recommended to make impression more innovative. I may suggest use of an emerging concept, nanoarchitectonics (as post-nanotechnology concept, see https://pubs.rsc.org/en/content/articlelanding/2021/nh/d0nh00680g). For example, the title like ... Theoretical nanoarchitectonics of GaN nanowires for ultraviolet irradiation-dependent electromechanical properties ... may sound more innovative and clear.

Response 1: The title has been revised as recommended.

Point 2: In conclusion, instead of only fact lists, more description including future perspectives had better be added.

Response 2: Following the suggestion, a detail description has been added, including the future perspectives (see par. 3 on page 8).

Point 3: Comparisons over the past approach have to be made to emphasize novelty and superiority of the current approach

Response 3: As suggested, we have provided the comparison over the past approach and highlighted the novelty and superiority of our approach (see par. 3 on page 1 and par. 1 on page 2).

Reviewer 3 Report

The manuscript entitled "Ultraviolet irradiation-dependent electromechanical properties of GaN nanowires" presents some interesting findings and can be further considered for review in Materials if the authors provide more data and comment on some of my questions.

First of all, from my point of view, the provided data is not sufficient to support the conclusions. Indeed, the results of the COMSOL modeling without experimental verification are only hypothetical data. The authors should connect their findings with reliable data in the literature.

Also, the authors should comment on their statement:

"Generally speaking, an analytic solution of such a nonlinear model is difficult or even impossible to be obtained. Hence, to solve the photoexcitation physical problem, a numerical iterative method is adopted by using the PDE module of COMSOL Multiphysics software."

How reliable the provided solution would be and any accuracy issues?

Author Response

Response to Reviewer 3 Comments

Point 1: First of all, from my point of view, the provided data is not sufficient to support the conclusions. Indeed, the results of the COMSOL modeling without experimental verification are only hypothetical data. The authors should connect their findings with reliable data in the literature. Also, the authors should comment on their statement: "Generally speaking, an analytic solution of such a nonlinear model is difficult or even impossible to be obtained. Hence, to solve the photoexcitation physical problem, a numerical iterative method is adopted by using the PDE module of COMSOL Multiphysics software. "How reliable the provided solution would be and any accuracy issues?

Response 1: Following the comment, it is worth noting that, as a preliminary study and for simplification, the effects of ultraviolet excitation on the electromechanical properties of nanowires were studied by using theoretical and numerical methods in this paper. Just as mentioned, the results of COMSOL modeling were obtained here without experimental verification. Further study is needed in order to completely understand the effect of such a complex and coupling problem. Regarding the calculation reliability and accuracy with the PDE module of COMSOL Multiphysics, it can be verified by using a perturbation method. This has been discussed and clarified in the revised version (see par. 2 on page 5).

Round 2

Reviewer 1 Report

The authors have addressed most of my comments and the manuscript could be accepted for publication.

Author Response

Response to Reviewer 1 Comments

Point 1: The authors have addressed most of my comments and the manuscript could be accepted for publication.

Response 1: Thanks a lot for your kind suggestion.

Reviewer 3 Report

The answers from the authors were relevant. The provided changes were sufficient. Please add the outlook/next steps in the end and please draw the most critical things to be investigated experimentally in future works.

Author Response

Response to Reviewer 3 Comments

Point 1: The answers from the authors were relevant. The provided changes were sufficient. Please add the outlook/next steps in the end and please draw the most critical things to be investigated experimentally in future work.

Response 1: Many thanks. As suggested, we have added the outlook and next-step works in the end of the newly revised version (see par. 7 on page 8 and par. 1 on page 9).
